# Anxiety, Depression, and Quality of Sleep Vary in Their Correlation to Postoperative Outcomes of Rotator Cuff Repair: A Prospective Study

**DOI:** 10.3390/jcm13113340

**Published:** 2024-06-05

**Authors:** Umile Giuseppe Longo, Martina Marino, Vincenzo Candela, Alessandra Greco, Ilaria Piergentili, Claudia Arias, Alessandro de Sire, Pieter D’Hooghe

**Affiliations:** 1Fondazione Policlinico Universitario Campus Bio-Medico, Via Alvaro del Portillo, 200, 00128 Rome, Italy; martinamarino.dr@gmail.com (M.M.); v.candela@policlinicocampus.it (V.C.); alessandra.greco5@gmail.com (A.G.); 2Research Unit of Orthopaedic and Trauma Surgery, Department of Medicine and Surgery, Università Campus Bio-Medico di Roma, Via Alvaro del Portillo, 21, 00128 Rome, Italy; 3CNR-IASI, Laboratorio di Biomatematica, Consiglio Nazionale delle Ricerche, Istituto di Analisi dei Sistemi ed Informatica, 00128 Rome, Italy; ilaria.piergentili94@gmail.com; 4Hospital Nacional Edgardo Rebagliati Martins, Lima 15072, Peru; claudia_arias81@hotmail.com; 5Physical Medicine and Rehabilitation Unit, Department of Medical and Surgical Sciences, University of Catanzaro “Magna Graecia”, 88100 Catanzaro, Italy; alessandro.desire@unicz.it; 6Research Center on Musculoskeletal Health, MusculoSkeletalHealth@UMG, University of Catanzaro “Magna Graecia”, 88100 Catanzaro, Italy; 7Aspetar Orthopedic and Sports Medicine Hospital, Aspire Zone, 1, Sportscity Street, Doha P.O. Box 29222, Qatar; pieter.orthopedie@gmail.com

**Keywords:** rotator cuff repair, anxiety, depression, quality of sleep

## Abstract

**Background/Objectives:** Recent studies imply that psychological factors and sleep quality play a role in the outcomes of surgical procedures, including in orthopedic surgery. The aim of the present study is to evaluate possible correlations between preoperative depression, anxiety, and quality of sleep and functional 6-month postoperative scores in patients having undergone rotator cuff repair (RCR). **Methods:** All patients included in the study performed the Hospital Anxiety and Depression Scale (HADS) and Pittsburgh Sleep Quality Index (PSQI) questionnaires preoperatively and 36-item Short-Form Health Survey (SF-36), Oxford Shoulder Score (OSS), Shoulder Pain and Disability Index (SPADI), and PSQI questionnaires at the six-month postoperative follow-up. A total of 47 patients were included in the analysis. **Results:** Statistically significant differences between preoperative anxious and not-anxious groups were found in the postoperative SF-36 Physical Component Summary (PCS) and Mental Component Summary (MCS) scores and PSQI score. The correlation of the preoperative depression score to postoperative outcome measures revealed a strong positive correlation between the preoperative HADS-D score and the 6-month PCS, MCS, and OSS scores. The correlation of preoperative sleep quality to postoperative outcome measures revealed a strong positive correlation between the preoperative PSQI score and 6-month MCS score. **Conclusions:** Anxious patients had worse postoperative RCR outcomes. Depression may be influenced by factors related to RC pathology; however, there were no statistically significant correlations. Sleep quality generally improves postoperatively, and no significant association was found between bad preoperative sleepers and worse outcomes.

## 1. Introduction

Rotator cuff pathology has an incidence of 16–34% in the general population and it is one of the most common causes of disability and pain related to the shoulder [1,2]. Incidence increases linearly with age starting from the third decade, and by the eighth decade, it affects about 50% of the general population [1,2]. The procedure of choice for management of rotator cuff tears, often following the failure of conservative treatment, is arthroscopic repair or mini-open repair [3,4,5,6,7]. An increase in indications for surgery in the United States and an increased incidence of the procedure world-wide have been reported [8,9,10]. However, despite the common use of this procedure, its success rates vary [11]. 

Recent studies have implied that psychological factors and sleep quality play a role in the outcomes of surgical procedures, including in the field of orthopedic surgery [12,13]. Concerns are directed towards patients suffering from anxiety and depression; in some studies, those suffering from the latter have responded poorly to surgical intervention [12,14]. However, despite such implications, other studies investigating the clinical outcomes of rotator cuff repair (RCR) have demonstrated that, although anxiety and depression correlate with preoperative pain and function, they are not necessarily a prediction of poor surgical outcomes [12,14].

In addition to psychological conditions, poor sleep quality is closely associated with rotator cuff pathology and RCR. Perhaps a good demonstration of this is that bad sleep quality is one of the driving motivations for which patients seek treatment [15]. Furthermore, good quality of sleep is believed to be closely associated with postoperative healing and increased overall patient satisfaction [15].

Some studies have highlighted the correlations between physiological factors and sleep quality with postoperative measures such as range of motion (RoM) or strength [16,17]. However, the data remain discordant regarding the correlation between psychological factors and sleep quality with functional postoperative outcomes. Therefore, the present study aims to evaluate possible correlations between preoperative depression, anxiety, and quality of sleep and functional 6-month postoperative scores in patients having undergone RCR. We hypothesize that the presence of depression, anxiety, or poor sleep quality in RCR patients preoperatively may negatively influence postoperative outcome scores.

## 2. Materials and Methods

The present study included patients with rotator cuff tears who underwent arthroscopic repair between January 2018 and January 2019 in our center. Ethics board approval was granted by the Institutional Review Board of Campus Bio-Medico University of Rome (COSMO study, protocol number: 78/18 OSS ComEt CBM, 16/10/18) and followed the guidelines of The Declaration of Helsinki.

Preoperative 1.5 T magnetic resonance imaging (MRI) and clinical evaluation by two orthopedic surgeons specialized in shoulder arthroscopy were used to examine the tears. Only patients with Goutallier grade 2 [18] and Patte stage 2 lesions [19] were included in this study. Ellman grade 2 tears with a size of 1–3 cm were included in the study, while lesions above 3 cm were not considered. All patients had previously received conservative treatment, including physical therapy and corticosteroid injections. Patients who were not scheduled to receive surgery or had other shoulder diseases were not included in the study. Patients suffering from rheumatoid arthritis and associated health conditions were also not included in the present study, as prednisone treatment may interfere with tissue and bone quality. 

A standardized rehabilitation protocol was followed by all patients [20]. The affected arm was supported using a sling equipped with an abduction pillow for a duration of six weeks. Patients were allowed active elbow flexion and extension, but terminal extension was restricted. Passive external rotation exercises were initiated from the first day after surgery and maintained within a comfortable range. Overhead stretching was intentionally limited until six weeks postoperatively to prevent any damage to the repair. At the six-week mark, the sling was removed, and patients began overhead stretching using a rope and pulley system. Isometric strengthening and rehabilitation exercises targeting the rotator cuff, deltoid, and scapular stabilizers were introduced 10 to 12 weeks after the operation. Rehabilitation efforts continued for a total of six months. Once shoulder strength was adequately restored, heavy manual work and overhead activities were permitted, typically occurring between 6 and 10 months after the surgery.

The study was performed on consecutive patients undergoing RCR for 12 months in the Orthopaedic Department of Campus Bio-Medico University Hospital of Rome. In addition, patients with at least six months of follow-up were included. This timeframe allowed sufficient time for recovery following the procedure. Additionally, 6-month follow-up was chosen given that, in previous studies, the normalization of depressive and anxious symptoms related to orthopedic conditions occurred at this time [21,22].

Preoperative questionnaires were administered by trained nursing staff on the day of intervention. Postoperative questionnaires were administered by the same nursing staff during the 6-month follow-up visit, or, alternatively, patients were contacted via telephone. All nursing staff received training for the administration of the questionnaires and were available to patients for clarification when necessary. Patients were not made aware of the goal of the study when filling out the questionnaires.

All patients included performed the Hospital Anxiety and Depression Scale (HADS) and Pittsburgh Sleep Quality Index (PSQI) questionnaire preoperatively and the 36-item Short-Form Health Survey (SF-36), HADS, Oxford Shoulder Score (OSS), Shoulder Pain and Disability Index (SPADI), and PSQI questionnaires at the 6-month postoperative follow-up. To facilitate the comparison, all questionnaire results were normalized with scores between 0 (best condition) and 100 (worse condition).

### 2.1. Preoperative Anxiety and Depression

Before surgery, preoperative anxiety and depression were evaluated with the HADS questionnaire in the validated Italian form [23]. The HADS questionnaire includes 7 items assessing anxiety (HADS-A) and 7 items assessing depression (HADS-D). Each dimension ranged between 0 (best score) and 21 (worse score) [24]. A value of HADS-A or HADS-D less than or equal to 7 corresponded to not having anxiety or depression. Values ranging from 8 to 10 corresponded to having low anxiety or depression. Values of more than or equal to 11 corresponded to having high anxiety or depression [24]. 

Therefore, patients with a HADS-A score of less than 11 were identified as not anxious and patients with HADS-D score of less than 11 were identified as not depressed, while patients with scores which were more than or equal to 11 were included in the anxiety and depression groups, respectively.

### 2.2. Preoperative Sleep Quality

The validated Italian form of the PSQI questionnaire [25] was used to assess the preoperative quality of sleep in patients. The PSQI is a questionnaire with 19 items, and it is used to assess sleep quality in patients via a self-reporting system [26]. The questionnaire evaluates seven sleep domains (sleep quality, latency, length, habitual sleep efficiency, sleep disruptions, sleep medication usage, and daytime dysfunction) to determine total sleep quality [25]. The PSQI is scored using seven component scores, each ranging from 0 (no difficulty) to 3 (extreme difficulty). The component scores were added together to create a total score (ranging from 0 to 21). 

Lower ratings imply greater sleep quality [27]. A PSQI score of less than 5.5 indicates good sleep [27]; therefore, we considered patients with PSQI values of less than 5.5 as good sleepers and patients with PSQI values of more than or equal to 5.5 as bad sleepers.

### 2.3. Postoperative Scores

At the six-month postoperative follow-up, the following questionnaires were administered: SF-36, HADS, OSS, SPADI, and PSQI.

### 2.4. Thirty-Six-Item Short-Form Health Survey 

The SF-36 score was validated in the Italian language [28]. This questionnaire is intended to provide measures of general health. The questionnaire comprises 36 items that generate eight dimensions (Physical Functioning, Social Functioning, Role Limitations caused by Physical Problems, Role Limitations caused by Emotional Problems, General Mental Health, Vitality, General Health Perceptions, and Pain) [28]. These eight dimensions are grouped into two components, the Physical Component Summary (PCS) and Mental Component Summary (MCS). In addition, a single unscaled question on health changes in the previous year was also included (Health Change) [28].

### 2.5. Oxford Shoulder Score

The validated Italian OSS score [29] was used. The OSS is a 12-item questionnaire for patients with shoulder degeneration or inflammation. Each question includes five response options, each corresponding to a score between 1 and 5. In addition, a score of 12 (best condition) to 60 (worst condition) is given. Both pain and its impact on daily life are investigated in this questionnaire [29].

### 2.6. Shoulder Pain and Disability Index

The SPADI score was validated in the Italian language and was created to assess the severity of pain and impairment caused by shoulder disease [30]. The SPADI is a self-administered index of 13 questions separated into two subscales, pain and disability, with a 5-item pain subscale and an 8-item disability subscale. Each subscale is added together and converted into a score of 100, with a higher score indicating more pain or disability [30].

### 2.7. Statistical Analysis

A priori power analysis was performed using a correlation of −0.626 between the OSS and preoperative HADS [12]. With a desired power level of 0.8 and a 0.05 level of significance (2-tailed), the minimum total sample size amounted to 17 subjects.

The normal distribution of the data was analyzed with the Shapiro–Wilk test. Since the data were abnormal, the differences in the scores between the groups (anxiety vs. no anxiety, depression vs. no depression, and good sleep vs. bad sleep) were calculated using the Independent-Samples Mann–Whitney U Test. The correlations between the preoperative scores (HADS-A, HADS-D, and PSQI) and the postoperative scores with Spearman’s correlation were also calculated.

All statistical assessments were performed using SPSS for Windows (version 26; IBM Corp., Armonk, NY, USA) and R software version i.386.4.0.3.

## 3. Results

A total of 47 patients were eligible for inclusion in the present trial and completed the preoperative and 6-month postoperative follow-up surveys; all of them were included in the study. The patients’ ages varied from 18 to 87 years, with 18 (38.3%) females and 29 (61.7%) males. 

### 3.1. Preoperative Anxiety

Among the 47 patients, 12 qualified as anxious (25.5%) and 35 as not anxious (74.52%). The median preoperative HADS-A score among anxious patients was 50 (range: 28.6 to 61.9), and it was 85.7 (range: 66.7 to 100) among those patients which were not anxious (*p* < 0.001). 

The correlation between the preoperative anxiety score and postoperative outcome measures (Table 1) revealed a strong positive correlation (rho = 0.826, *p* < 0.001) between the preoperative HADS-A score and 6-month MCS score. There was a high–medium positive correlation between the HADS-A score and 6-month PCS (rho = 0.425 and *p* = 0.003), SPADI Disability (rho = 0.402 and *p* = 0.005), and PSQI (rho = 0.486 and *p* = 0.001) scores. There was a low–medium positive correlation (rho = 0.350, *p* = 0.016) between the preoperative HADS-A score and the 6-month SPADI Pain score.

Statistically significant differences between preoperative anxious and not-anxious groups were found in the postoperative PCS score, MCS score, and PSQI score (Table 2). The median value of PCS in anxious patients was 73.4 (range: 28.1 to 97.5), and in patients which were not anxious, it was 90 (range: 48.8 to 98.8; *p* = 0.036). The median value of MCS in anxious patients was 73.9 (range: 28.6 to 93.3), and in patients which were not anxious, it was 91.4 (range: 73 to 98.8; *p* = 0.001). The median value of PSQI in anxious patients was 69 (range: 42.9 to 81), and in patients which were not anxious, was 76.2 (range: 57.1 to 100; *p* = 0.002).

The radar plot in Figure 1 shows the differences between the groups.

### 3.2. Preoperative Depression

Among the 47 patients, 6 qualified as depressed (12.8%) and 41 as not depressed (87.2%). The median preoperative HADS-D score among depressed patients was 57.1 (range: 19 to 61.9), and it was 90.5 (range: 66.7 to 100) among those patients which were not depressed (*p* < 0.001). 

The correlation of preoperative depression score to postoperative outcome measures (Table 1) revealed a strong positive correlation between the preoperative HADS-D score and 6-month PCS (rho = 0.574, *p* < 0.001), MCS (rho = 0.556, *p* < 0.001), and OSS (rho = 0.504, *p* < 0.001) scores. There was a high–medium positive correlation between the HADS-D score and 6-month SPADI pain (rho = 0.397 and *p* = 0.006) and SPADI Disability (rho = 0.413 and *p* = 0.004) scores. There was a low positive correlation (rho = 0.309, *p* = 0.035) between the preoperative HADS-D score and the 6-month PSQI score.

No statistically significant differences between preoperative depressed patients and patients which were not depressed were found in postoperative outcomes (Table 3, Figure 2).

### 3.3. Preoperative Quality of Sleep

Among the 47 patients, 43 qualified as bad sleepers (91.5%) and 4 qualified as good sleepers (8.5%). The median preoperative PSQI score among the bad sleepers was 61.9 (range: 14.3 to 71.4), and it was 81 (range: 76.2 to 90.5) among the good sleepers (*p* < 0.001). 

The correlation of the preoperative sleep quality score to the postoperative outcome measures (Table 4) revealed a strong positive correlation between the preoperative PSQI score and 6-month MCS (rho = 0.503, *p* < 0.001) score. There was a high–medium positive correlation between the PSQI score and 6-month PCS (rho = 0.472 and *p* = 0.001) and HADS-A (rho = 0.486, *p* = 0.001) scores. There was a low positive correlation between preoperative PSQI score and 6-month HADS-D (rho = 0.309, *p* = 0.035) and SPADI Disability (rho = 0.294, *p* = 0.045) scores.

No statistically significant differences between the preoperative bad-sleeper and good-sleeper groups were found in the postoperative outcomes (Table 5, Figure 3).

## 4. Discussion

The present study evaluates the effects of sleep quality, anxiety, and depression on the postoperative outcomes of RCR patients. Statistically significant differences in postoperative outcome scores were found in the anxious compared to non-anxious groups. A strong positive correlation was found between the preoperative HADS-D and postoperative PCS, MCS, and OSS scores. Finally, a strong positive correlation was found between the preoperative PSQI and postoperative MCS score. 

Strong positive correlation was found between the preoperative HADS-A score and 6-month postoperative MCS score. A high–medium positive correlation was found between the HADS-A and 6-month PCS, SPADI disability, and PSQI scores. While a low–medium positive correlation was found between the HADS-A and SPADI pain scores. Such correlations suggest that patients suffering from higher levels of anxiety generally will have improvement in postoperative outcome scores. Furthermore, the strong positive correlation between the HADS-A score and MCS scores suggests that anxiety may improve after surgery. These data support the hypothesis that patients may classify as preoperatively anxious due to the emotional state experienced before undergoing a surgical procedure rather than the injury itself [31,32]. Several studies have demonstrated that hospitalization is a source of anxiety for many patients, and that the associated stress response negatively impacts recovery [31,32]. For this reason, some authors suggest disclosing to patients that anxiety levels decrease postoperatively given the high success rate of the procedure during preoperative counseling [12,32]. Additional solutions may include out-patient one-day care surgeries to assist in decreasing anxiety levels; on the one hand, this approach may be cost-effective, but, on the other hand, patient selection must be carefully carried out to avoid possible complications or readmissions [33,34].

Differences in the outcomes of surgery, measured via PCS, MCS, and PSQI, were found to be statistically significant in the anxious and not-anxious groups, suggesting that anxiety does play a negative role in postoperative outcomes (Table 2). Other published studies revealed that high preoperative anxiety increased pain scores over 24 h postoperatively, as well as causing an increase in analgesic consumption [32,35]. Another systematic review concluded that anxiety may explain the adverse effects of certain personality traits on postoperative outcomes [36]. These findings combined suggest that anxiety management preoperatively could be worthwhile for the improvement of surgical outcomes. Some studies suggest that preoperative counseling may significantly reduce anxiety and improve postoperative outcomes, as well as consequentially decrease analgesic consumption [32,36]. Despite these conclusions, more studies are needed to confirm the relationship between anxiety and postoperative outcomes.

Only 6 of the 47 included patients were classified as depressed preoperatively, and this value may indicate that preoperative depression is not directly caused by hospitalization. A high to medium–high correlation has been found between preoperative HADS-D and PCS, MCS, OSS, SPADI Pain, and SPADI Disability. These results suggest that, in depressed patients, RC pathology was a contributor to the depressive state, likely due to its consequences on quality of life, as seen by the overall improved postoperative scores. However, given that very few patients suffer from these symptoms, pre-existing risk factors for depression could play a role in preoperative and postoperative depression. No statistically significant postoperative outcome difference was found between the depressed and not-depressed groups.

It is relevant to consider anxiety and depression as inter-related conditions. A study reports difficulties in distinguishing between anxiety and depression, and highlights that these two constructs are deeply connected [37]. In the present study, the analysis of both preoperative depression and anxiety revealed a high to medium–high correlation with the MCS and PCS postoperative values, suggesting similar correlations to postoperative outcomes. A very recent study assessing the impact of psychological well-being on patients who underwent rotator cuff repair found that a lower perceived quality of life was associated with worse functional outcomes, anxiety, and depression symptoms as well as pain following RCR [38]. These findings support the hypothesis that worsened quality of life due to rotator cuff pathology and postoperative outcomes following RCR may have mutual effects on one another. Further targeted research on this topic is required to elucidate these specific correlations.

Most patients in the study qualified as bad sleepers, and a strong or high–medium correlation between preoperative scores and MCS, PCS, and HADS-A was found. Of particular interest is the correlation between the preoperative scores and HADS-A, highlighting a possible connection between the quality of sleep of patients and anxiety levels. The findings of a systematic review exploring sleep disturbances and rotator cuff tears reveal that other factors, including comorbidities, psychiatric disturbances, and narcotic consumption, contribute to worsened quality of sleep [39]. Although there may be a correlation with improved anxiety levels postoperatively, a low correlation was found between preoperative scores and depression, assessed thanks to the HADS-D subscale. A low correlation was also found between preoperative scores and the disability caused by the condition assessed via the SPADI Disability scale. Although poor sleep quality is prevalent preoperatively, studies demonstrate that sleep quality improves progressively after surgery [39,40].

Poor sleep quality has been shown to play a role in anxiety and depression and is a common complaint in many shoulder disorders [41,42]. Although past studies have definitively shown that mental health disorders alter the quality of sleep, recently it has come to light that bad sleep quality may worsen mental health symptoms such as anxiety and depression [42]. This suggests that anxiety and sleep mutually influence one another. Such observations provide a better understanding of the correlation between sleep quality and [14] anxiety preoperatively. However, no difference in postoperative outcomes was found between good and bad sleepers; nevertheless, more studies with larger patient cohorts may be necessary to better evaluate such a correlation. 

Of important note is that the current study is similar in terms of methodology and aim to a previously published work [14] by the current authors. The time of recruitment between the two studies was not conducted during overlapping timeframes, and thus patient cohorts are distinct from one another. The current study additionally includes the analysis of sleep quality, which was a parameter that was not assessed in the prior study. Considering such information, the current investigation and its results remain relevant, and allow for further insight into this topic, which remains under-investigated in the literature. 

The limitations of the present study include the use of patient-reported outcome measures, which may influence the result comparability. This is because the use of these measures inevitably introduces a level of patient subjectivity to the data collection process. To minimize variation in the data, surveys were translated professionally into the Italian language, and trained nurses supervised the process in case of any patient confusion and were trained to provide necessary clarification. The present study considered for inclusion all patients that underwent RCR in our center over one year, leading to a cohort of 47 patients. Unfortunately, practical constraints influenced the choice to limit recruitment to one year, but such a decision allowed us to balance scientific rigor with feasibility, maintaining the study’s integrity. While a larger cohort would have been beneficial to the present study, statistical significance was achieved nonetheless. Furthermore, in future explorations, the stratification of results for age and sex may be useful to further analyze the effects of anxiety, depression, and sleep quality on different patient groups. Additionally, considering the education levels of patients alone and in concomitance with the quality of perioperative information provided could provide insights into patients’ ability to understand and manage postoperative care and in the evaluation of the quality of the educational tools provided. The presence or absence of handouts for postoperative care and patient understanding of expected timelines in postoperative recovery could also be examined. Assessing these factors can offer a comprehensive view on how well patients are equipped to handle their own recovery process, and how surgeons and medical professionals could improve patient education, both of which may contribute to improving or harming patients’ psychosocial state. Another factor which was not considered in the present study is the financial burden associated with time away from work during recovery; taking this parameter into account could also provide an understanding on the broader impacts on patients’ overall well-being.

## 5. Conclusions

The correlations between preoperative and postoperative anxiety, depression, and sleep quality are diverse. The findings reveal that anxious patients had worse postoperative RCR outcomes than those in the not-anxious group. These results highlight the need for more studies evaluating the influence of anxiety on orthopedic surgical outcomes, and perhaps regarding the effects of preoperative psychological counseling. Only a small subset of patients was found to experience a preoperative depressive state. The results suggest that depression may be influenced by factors that alter quality of life due to RC pathology. However, there was no statistically significant difference in the outcomes between the preoperatively depressed and not-depressed groups. Most patients in the study qualified as bad sleepers, and the data showed a correlation between preoperative bad sleep and anxiety. However, the data also suggest that sleep quality improves postoperatively, while no significant association was found between bad preoperative sleepers and worse postoperative outcomes. More studies investigating the correlations between psychiatric conditions and sleep quality on the postoperative outcomes of orthopedic procedures is strongly encouraged. A better understanding of the interplay between these factors may lead to better patient care via the development of more targeted pre- and postoperative counseling programs.

## Figures and Tables

**Figure 1 jcm-13-03340-f001:**
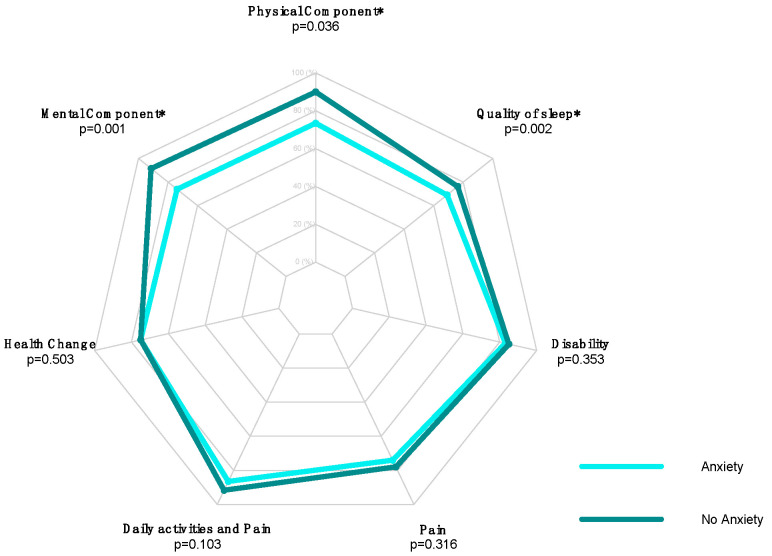
Radar plot comparing postoperative SF-36 Health Change, SF-36 MCS, SF-36 PCS, PSQI, SPADI Disability, SPADI Pain, and Daily Activities and Pain scores in the “anxiety” versus “no anxiety” groups of RCR patients (* = Statistically Significant Value).

**Figure 2 jcm-13-03340-f002:**
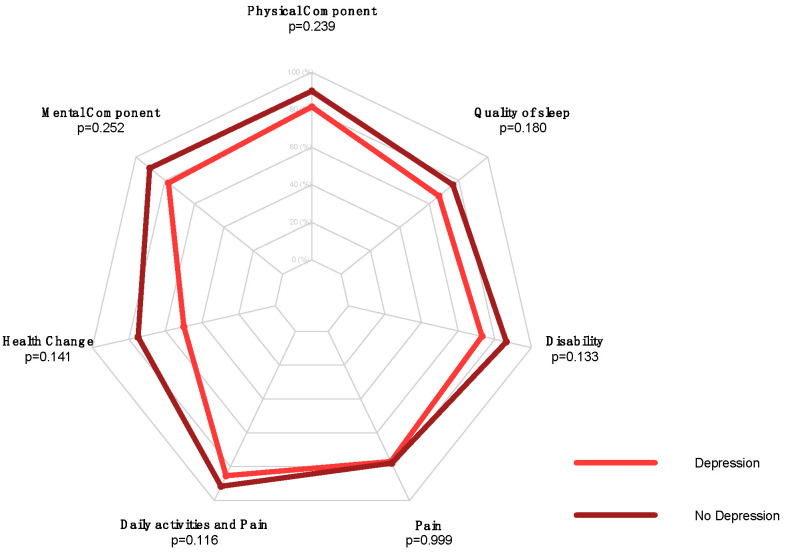
Radar plot comparing postoperative SF-36 Health Change, SF-36 MCS, SF-36 PCS, PSQI, SPADI Disability, SPADI Pain, and Daily Activities and Pain scores in “depression” versus “no depression” groups of RCR patients.

**Figure 3 jcm-13-03340-f003:**
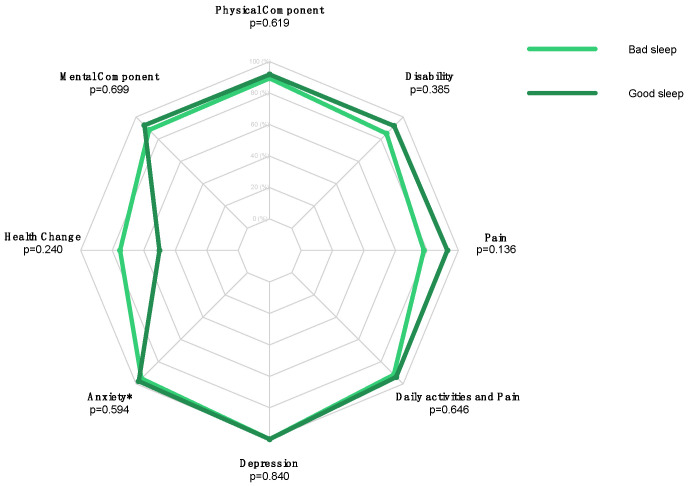
Radar plot comparing postoperative SF-36 Health Change, SF-36 MCS, SF-36 PCS, SPADI Disability, SPADI Pain, Daily Activities and Pain, HADS-D, and HADS-A scores in the “bad sleep” versus “good sleep” groups of RCR patients. (* = Statistically Significant Value).

**Table 1 jcm-13-03340-t001:** Correlations between the preoperative HADS and postoperative scores.

Parameter	HADS-A		HADS-D	
	rho	*p*-Value	rho	*p*-Value
SF-36 PCS	0.425	0.003 *	0.574	<0.001 *
SF-36 MCS	0.826	<0.001 *	0.556	<0.001 *
SF-36 Health Change	0.119	0.426	−0.012	0.938
OSS	0.257	0.081	0.504	<0.001 *
SPADI PAIN	0.350	0.016 *	0.397	0.006 *
SPADI DISABILITY	0.402	0.005 *	0.413	0.004 *
PSQI	0.486	0.001 *	0.309	0.035 *

HADS-A: Hospital Anxiety and Depression Scale–Anxiety; HADS-D: Hospital Anxiety and Depression Scale–Depression; SF-36 PCS: 36-item Short-Form Health Survey–Physical Component Summary; MCS: Mental Component Summary; OSS: Oxford Shoulder Score; SPADI: Shoulder Pain and Disability Index; PSQI: Pittsburgh Sleep Quality Index. (* = Statistically Significant Value).

**Table 2 jcm-13-03340-t002:** Median (min–max) values of the postoperative HADS-A scores.

Parameter	Anxiety (N = 12)		No Anxiety (N = 35)		*p*-Value
	Median	Range	Median	Range	
SF-36 PCS	73.4	28.1–97.5	90	48.8–98.8	0.036 *
SF-36 MCS	73.9	28.6–93.3	91.4	73.0–98.8	0.001 *
SF-36 Health Change	75	0.0–100.0	75	50.0–100.0	0.503
OSS	86.5	58.3–97.9	91.7	50.0–100.0	0.103
SPADI PAIN	74	46.0–94.0	78	46.0–100.0	0.316
SPADI DISABILITY	83.1	45.0–96.3	85	58.8–100.0	0.353
PSQI	69	42.9–81.0	76.2	57.1–100.0	0.002 *

HADS-A: Hospital Anxiety and Depression Scale–Anxiety; SF-36 PCS: 36-item Short-Form Health Survey–Physical Component Summary; MCS: Mental Component Summary; OSS: Oxford Shoulder Score; SPADI: Shoulder Pain and Disability Index; PSQI: Pittsburgh Sleep Quality Index. (* = Statistically Significant Value).

**Table 3 jcm-13-03340-t003:** Median (min–max) values of the postoperative HADS-D scores.

Parameter	Depression (N = 6)		No Depression (N = 41)		*p*-Value
	Median	Range	Median	Range	
SF-36 PCS	81.6	28.1–98.8	90	36.3–98.8	0.239
SF-36 MCS	77.8	28.6–96.5	90.5	36.3–98.8	0.252
SF-36 Health Change	50	0.0–75.0	75	50.0–100.0	0.141
OSS	85.4	58.3–95.8	91.7	50.0–100.0	0.116
SPADI PAIN	77	46.0–94.0	78	46.0–100.0	0.999
SPADI DISABILITY	73.1	45.0–96.3	86.3	58.8–100.0	0.133
PSQI	66.7	42.9–100.0	76.2	52.4–95.2	0.180

HADS-D: Hospital Anxiety and Depression Scale–Depression; SF-36 PCS: 36-item Short-Form Health Survey–Physical Component Summary; MCS: Mental Component Summary; OSS: Oxford Shoulder Score; SPADI: Shoulder Pain and Disability Index; PSQI: Pittsburgh Sleep Quality Index.

**Table 4 jcm-13-03340-t004:** Correlations between the values of the preoperative PSQI and postoperative scores.

Parameter	PSQI	
	rho	*p*-Value
SF-36 PCS	0.472	0.001 *
SF-36 MCS	0.503	<0.001 *
SF-36 Health Change	0.05	0.736
HADS-A	0.486	0.001 *
HADS-D	0.309	0.035 *
OSS	0.275	0.061
SPADI PAIN	0.272	0.064
SPADI DISABILITY	0.294	0.045 *

SF-36 PCS: 36-item Short-Form Health Survey–Physical Component Summary; MCS: Mental Component Summary; HADS-A: Hospital Anxiety and Depression Scale–Anxiety; HADS-D: Hospital Anxiety and Depression Scale–Depression; OSS: Oxford Shoulder Score; SPADI: Shoulder Pain and Disability Index. (* = Statistically Significant Value).

**Table 5 jcm-13-03340-t005:** Median (min–max) values of the PSQI score.

Parameter	Bad Sleep (N = 43)		Good Sleep (N = 4)		*p*-Value
	Median	Range	Median	Range	
SF-36 PCS	89.4	28.1–98.8	91.9	50.0–96.3	0.619
SF-36 MCS	88.3	28.6–98.8	92.5	74.0–93.3	0.699
SF-36 Health Change	75	0.0–100.0	50	50.0–75.0	0.240
HADS-A	95.2	38.1–100.0	97.6	81.0–100.0	0.594
HADS-D	100	23.8–100.0	100	85.7–100.0	0.840
OSS	91.7	58.3–97.9	93.8	50.0–100.0	0.646
SPADI PAIN	78	46.0–100.0	93	52.0–94.0	0.136
SPADI DISABILITY	85	45.0–100.0	91.9	66.3–96.3	0.385

PSQI: Pittsburgh Sleep Quality Index; SF-36 PCS: 36-item Short-Form Health Survey–Physical Component Summary; MCS: Mental Component Summary; HADS-A: Hospital Anxiety and Depression Scale–Anxiety; HADS-D: Hospital Anxiety and Depression Scale–Depression; OSS: Oxford Shoulder Score; SPADI: Shoulder Pain and Disability Index.

## Data Availability

The datasets used and/or analyzed during the current study are available from the corresponding author on reasonable request.

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
