# Peer review of "Anxiety, Depression, and Quality of Sleep Vary in Their Correlation to Postoperative Outcomes of Rotator Cuff Repair: A Prospective Study"

_jcm, 2024, doi:10.3390/jcm13113340_

Round 1

Reviewer 1 Report

Comments and Suggestions for Authors

Dear Authors,

I was quite interested to read your manuscript as most of my clinical practice focuses on rotator cuff pathology. Therefore, your study is applicable to many of my own patients.

The topic is interesting as encountered regularly in our clinics but not studied extensively. I thought the manuscript was overall well written and easy to read. In addition, there is potential to expand such results to numerous conditions treated in orthopaedic surgery.

A few things need to be refined all along your manuscript. The "Materials and Methods" section is where I had most of my questions. Please review all my comments, questions, suggestions in the attached PDF. Please review all the way down to the references section.

Having said that, I have one major concern regarding this work. One of your references is your own work on anxiety and depression after cuff repair that you published in the same journal last year (2023). The topic is very similar and so were the findings. I believe that in the current submission, you should have stated very clearly that this manuscript is "Part 2" or the sequel of this prior work. One could superficially state that the current submission is basically the same as the last publication on this topic (with similar scores reported at similar time frames), besides the sleep quality portion and dismiss your current work. But I believe that if you modify the current submission and ensure it is loud and clear (and well articulated) that this submission is Part 2 of prior work because of whatever you found in the prior work, the current submission would be more acceptable for publication. Otherwise, it really looks like you tried to get 2 papers of the same work and it does not look good.

Author Response

Thank you very much for your comments, the necessary changes were made to the various sections of the manuscript and highlighted according to the annotated PDF. We strongly believe that these changes have greatly improved the quality of our work.

The only comment which was not modified was the following “Please add the sample size calculation.”: We thank the reviewer for the comment, which has provided us with the opportunity to clarify this aspect. We have conducted a statistical consultation. In the statistical section, we have outlined the method of calculating the power analysis and sample size by stating: "A priori power analysis was performed using a correlation of -0.626 between OSS and preoperative HADS [12]. With a desired power level of 0.8 and a 0.05 level of significance (2-tailed), the minimum total sample size amounted to 17 subjects.” We believe we have accurately indicated this in this section. Provided that the editor disagrees with the representation detail of the statistics, we will be happy to change the methodology as suggested by the editor.

Additionally, in regards to the similar work published last year, we want to clarify that the cohorts are completely separate, and no patients overlap between the two, given that the time frame for data collection was different between the two studies. The following rationale was provided within the text: “Of important note is that the current study is similar in terms of methodology and aim to a previous published work [14] by the current authors. Time of recruitment between the two studies was not conducted during overlapping timeframes and thus patient cohorts are distinct from one another. The current study additionally includes the analysis of sleep quality, which was a parameter that was not assessed in the priori study. Considering such information the current investigation and its results remain relevant, and allow for further insight into this topic, which remains under-investigated in literature.”.

Reviewer 2 Report

Comments and Suggestions for Authors

The work is written interestingly. I congratulate the authors on their idea. In my opinion, several things need clarification. There is a lack of a research hypothesis. First and foremost, how the sample size was calculated. Calculations should be added. The group seems quite small.

Additionally, the authors write about the use of MRI but do not mention its power (how many Tesla MRI). How was the image processed and analyzed?

Furthermore, there is a lack of effect size calculations. The mere p-value calculations are insufficient. Additionally, please make editorial corrections. Use of font and its size should be in accordance with the journal's standards.

Author Response

Thank you very much for your comments, we have added the resolution of MRI imaging used and have included the research hypothesis in the new version of the manuscript. We also made sure that all editorial corrections were made appropriately, and that font size and style are in accordance with the journal’s standards.

We also thank the reviewer for the comments provided regarding the statistical analysis, more specifically the sample-size calculation, effect size calculations, and p-value calculations. These comments have provided us with the opportunity to clarify this aspect. We have conducted a statistical consultation. In the statistical section, we have outlined the method of calculating the power analysis and sample size by stating: "A priori power analysis was performed using a correlation of -0.626 between OSS and preoperative HADS [12]. With a desired power level of 0.8 and a 0.05 level of significance (2-tailed), the minimum total sample size amounted to 17 subjects. The normal distribution of the data was analysed with the Shapiro-Wilk test. Since the data was abnormal, the differences in the scores between the groups (anxiety vs no anxiety, depression vs no depression and good sleep vs bad sleep) were calculated using the Independent-Samples Mann-Whitney U Test. The correlations between the preoperative scores (HADS-A, HADS-D and PSQI) and the postoperative scores with Spearman’s correlation were also calculated.All statistical assessments were performed using SPSS for Windows (version 26; Armonk, NY: IBM Corp) and R software version i.386.4.0.3.” We believe we have accurately indicated this in this section. Provided that the editor disagrees with the representation detail of the statistics, we will be happy to change the methodology as suggested by the editor.

Round 2

Reviewer 1 Report

Comments and Suggestions for Authors

Dear Authors,

Thank you for sending a revised version of your manuscript. You answered my questions and revised your paper satisfactorily.

I believe this version is now ready for publication.

Author Response

Thank you. 

Reviewer 2 Report

Comments and Suggestions for Authors

The authors have adequately responded to my comments. I have no further remarks.

Author Response

Thank you.